# A Blockchain-Based Secure Image Encryption Scheme for the Industrial Internet of Things

**DOI:** 10.3390/e22020175

**Published:** 2020-02-04

**Authors:** Prince Waqas Khan, Yungcheol Byun

**Affiliations:** Department of Computer Engineering, Jeju National University, Jeju-si 690-011, Korea; princewaqas12@hotmail.com

**Keywords:** blockchain, image encryption, industrial Internet of Things, IIoT, security, privacy, image sensors, entropy analysis

## Abstract

Smart cameras and image sensors are widely used in industrial processes, from the designing to the quality checking of the final product. Images generated by these sensors are at continuous risk of disclosure and privacy breach in the industrial Internet of Things (IIoT). Traditional solutions to secure sensitive data fade in IIoT environments because of the involvement of third parties. Blockchain technology is the modern-day solution for trust issues and eliminating or minimizing the role of the third party. In the context of the IIoT, we propose a permissioned private blockchain-based solution to secure the image while encrypting it. In this scheme, the cryptographic pixel values of an image are stored on the blockchain, ensuring the privacy and security of the image data. Based on the number of pixels change rate (NPCR), the unified averaged changed intensity (UACI), and information entropy analysis, we evaluate the strength of proposed image encryption algorithm ciphers with respect to differential attacks. We obtained entropy values near to an ideal value of 8, which is considered to be safe from brute force attack. Encrypted results show that the proposed scheme is highly effective for data leakage prevention and security.

## 1. Introduction

Industry 4.0 is the fourth generation industrial revolution, which has grown exponentially from the Internet of Things (IoT) to the industrial IoT (IIoT). This innovation has provided solutions to the new challenges for the industrial sector. IoT allows connecting multiple devices at one time more conveniently without the need for human intervention [1]. IIoT is applied in the manufacturing process, supply chains, monitoring, and management systems. It deals with the connectivity of smart factories, machines, management systems, and other streamline business operations. It uses more sensitive and precise sensors than IoT, including more location-aware technologies on the supply chain side [2]. Nowadays, smart devices are used in different areas of industry, such as drones to monitor oil pipelines, sensors to monitor factories, water treating equipment, and tractors in agriculture. Smart cities could be a combination of commercial IoT and IIoT. The IIoT can massively increase connectivity and scalability and save time and costs for industrial organizations.

IIoT is the interjection of advanced machines and sensors in various industries, i.e., aerospace, health, energy, and defense. These are the systems whose malfunctioning often lead to life-threatening or other emergencies. Therefore, this sector requires intensive care and a high level of security. It is sweeping across industries, starting from the very primary manufacturing sector to high-level production units. It consists of production, business models, customer relations, research projects, education, and regional, national, and worldwide strategies of innovation [3]. The main motivations behind this improvement may be the quickly expanding digitization of the economy. Industry 4.0 depends on the number of innovative developments. It is a standout amongst continuously increasing data and correspondence technologies. These technologies need help and support to digitize the majority of the data. Theyrely upon decentralization, virtualization, interoperability, and ongoing capability. Various commercial enterprises around those realities are set on industry 4.0.

Image and video surveillance has a vital role in IIoT-oriented network computing systems. It is being used to build information management systems, business intelligence, scientific research, and real-time analysis [4]. Image sensor-based solutions help in optimizing maintenance and work process safety. In addition, they significantly improve the quality of productivity. Computer vision technology is used for the continuous monitoring and visual quality control of production processes. Increasingly advanced cameras and sensors are being used in the industry. This advancement also brings new challenges, such as the lack of built-in security. Establishing reliability, scalability, and IoT management in these sensors is complex. These new challenges demand safe and secure applications and devices without human intervention. Cyber-attacks against automation in the IIoT environment have far-reaching effects [5]. They can disrupt the manufacturing process, and sensitive data could be lost. They can have a negative impact on the quality of products. They can also damage property or harm humans.

Existing solutions for image encryption do not help smart industries where peers are decentralized. The history of blockchain technology (BCT) comes from cryptocurrency or Bitcoin. Blockchain is a ledger that records all the transactions that are sent between individuals [6]. This ledger is decentralized, so there is no central authority, like banks are in the non-digital world. It is also cryptographically secure and immutable. If something is added to this ledger, it cannot be deleted or changed. A blockchain is a consensus-based system, meaning that every node verifies the occurrence of the transaction and comes to a consensus about all the information before it is placed into the BCT-based ledger. BCT has emerged as a solution for security problems [7]. Due to its dynamic properties, industries have realized that it can play a critical role in the IIoT. This technology can be applied in the industry, not only from a security perspective but also from a transparency and regulatory compliments perspective. A blockchain can significantly revolutionize IIoT. It provides security, peer-to-peer device communications, and new functionality via smart contracts.

In this paper, an image encryption scheme for an IIoT-oriented network computing system is presented, which is based on a blockchain. We also apply different tests and compare our results with existing literature to prove the efficiency of our proposed work. The current work is discussed in Section 2. The main reason behind using a blockchain for the industrial environment and the critical features of the blockchain are highlighted in Section 3. The proposed encryption framework is summarized in Section 4. Section 5 explains the experimental results and compares them with existing work. Section 6 shines a light on possible relevant issues that could be faced during the implementation of our proposed scheme. We conclude our work in Section 7.

## 2. Related Works

The security and privacy in IIoT have been a hot topic for researchers since the evaluation of the industrial sector. We reviewed research articles that focus on preserving privacy in IoT and security vulnerabilities. We analyzed the literature regarding the use of a blockchain in IIoT and image encryption algorithms. An image encryption method based on a chaotic system was proposed by [8]. There are two kinds of encryption schemes. One is symmetric, and the other is asymmetric. In symmetric encryption, the encryption key is the same as the decoding key, so that it is not difficult to ascertain the other key that is dependent on one public key. Huang et al. [9] proposed a color image encryption framework in which the permutation–diffusion operation acts synchronously. More explicitly, the simultaneous permutation diffusion operation prompts communication among confusion and diffusion, and accordingly leads to low computational complexity and a high speed. In any case, one of the most noteworthy issues in symmetric encryption is the administration and dispersion of the key. The key may experience the risk of assailant interception in its distribution process. Asymmetric encryption necessitates that the encryption key be different from the decryption key and that the encryption key does not determine the decryption key. Chen et al. [10] proposed one such scheme utilizing a chaotic Ushiki map and the equal modulus decomposition of a color image in fractional Fourier space. The delicate starting values and disordered data could be considered as extra keys and a public key. Hyperspectral encryption processes utilizing 3D Arnold and gyrator transform have been associated with changing the pixel value of the encrypted image. In this modification process, pixel values will be revised and look jumbled accordingly, accomplishing image encryption. It broadens the image from spatial space to spectrum space. The strength of this cryptosystem is improved because secret information in the spectrum space is challenging to recover from the spatial space [11].

Another image encryption technique, which was based on the Choquet fuzzy integral and the hyperchaotic system, was presented in [12]. Visual data are usually present in large amounts, so different researchers have employed various methods of compressing the image data [13], and they have also presented summarization methods for videos and images [14]. He et al. [15] propose a strategy dependent on wavelength multiplexing diffraction imaging. They utilize classical encryption of an optical phase cover. It is exceptionally helpful for data storage and transmission. The decryption procedure is performed based on a mixed state decomposing algorithm. They exhibit an image encryption scheme dependent on a diffraction imaging of a blended state. They encode and consolidate color image layers, Red, Green, and Blue, in a solitary grayscale image. In the medical context, individual identity information is enormously important and personally related to a patient’s privacy. Some work has also been done to allow the medical field to protect the data regarding medical assets, medical surgery, surgical process management, and patient information management in a health center with the help of a radio frequency identification (RFID) application on the IoT [16]. There are two types of architecture in RFID systems: one is between the server and the wired reader, and other is between the server and the wireless reader, but the fixed reader can be shifted smoothly if required. In RFID-based medical health systems, RFID tags are attached to certain assets, such as management devices, and additional equipment [17]. The fixed reader can obtain valuable information about these objects and then communicate reliably with the server over a terminal computer. The data in the medical field from various insurance companies can also be encrypted. This data will be saved on a back-end server in the encryption order because many agencies such as insurance and cosmetic surgery agencies obtain information in the medical field from a third party. The result may be harmful to property and human health. The proposed scheme consumes fewer computing resources, meets the security requirement of ambiguity, deploys attack conflict synchronization, and forwards mutual security authentication and the non-denial of services. This scheme is a more capable system than the current medical system, and can secure medical field data privacy. The integration of blockchain and edge computing is also presented in [18]. They proposed a mechanism to store and manage data securely.

Some researchers have also used a blockchain for IoT [19]. One of them introduced a secure way of trading energy in IIoT using consortium blockchain. The challenges related to privacy and security in IIoT is discussed by [20] They briefly discussed attacks on IIoT systems and methods to secure them using security architectures and integrity verification. We also find security vulnerabilities in application systems of IoT in [21]. The authors explained insecure web interfaces, the lack of cryptographic supports, and SQL injections.

### Why Blockchain?

There are many existing techniques of image encryption, but none of them comply with the needs of the industry. Smart industries have decentralized networks of peers. Smart industries have many interconnected IoT devices, and they share sensitive data which is at continuous risk of exposure. The blockchain provides a complete solution for decentralized devices, and its encryption mechanism is very secure, which makes it perfect for smart industries. Table 1 provides a comparison of existing encryption solutions. Based on this comparison, we proposed a blockchain-based encryption method for sensitive images of smart industries.

IIoT will be based on the interconnected network of trusted devices, which will eradicate the need for mediators. With the advancements in the industrial manufacturing process, the concept of decentralization has become a trending idea. Decentralization often leads to more vulnerability and accessibility to hackers. There is a lack of accountability, and it is a significant obstacle in the trust issues regarding decentralization. A blockchain provides an immutable ledger, which keeps a record of all transactions, so it gives the surety that data is not tampered with. Another great feature that a blockchain provides is a smart contract, which is a self-executing piece of code for executing terms of agreement automatically. A blockchain has a significant impact on the product supply chain. It allows for maintaining product supply history. It keeps every participant of the chain up to date with knowledge of the product. It integrates different sections of an enterprise and allows them to communicate efficiently. A private blockchain such as a hyperledger fabric issues certificates to every user. It comes with a membership service provider (MSP) feature as well. An MSP defines different roles to each set of users and provides access privileges. The IIoT has heterogeneous devices, and every device generates and processes a vast amount of critical data, including sensitive images. Some images are only to be shared with particular participants within the IIoT environment. Some devices do not have enough computational power, so they need to offload their data, which raises an issue for the security of sensitive images. Most of the conventional security mechanisms are central; they are not suitable for the IIoT environment. A blockchain fulfills the demand for a decentralized and secure system.

## 3. Key Features of Blockchain

The blockchain was initially developed for monetary purposes. Now it is emerging from cryptocurrency and is likely to have a great impact across many industries. Its basic purpose was to eliminate third parties from money transactions by creating a trustworthy digital currency. A blockchain is a digital ledger that contains the entire history of transactions made on the network. It is an accumulation of linked blocks that are joined together by hash values that have been created over time. All information on the blockchain is permanent and cannot be changed. Figure 1 demonstrates a simpler version of blockchain. The very first block of this chain is called a genesis block [31]. Every node of this chain contains all the information, and it is linked with the hashed address of its previous node.

A hash identifies the block and all its content, and just like a human fingerprint, it is always unique. Once a block is created, its hash is being calculated. Changing something inside the block will cause the hash to change. Every block also contains the hash of the previous block. This effectively creates a chain of blocks. A blockchain is a peer-to-peer network, so it does not contain any central authority. Every node of blockchain receives a full copy of the whole chain, so nodes use that copy to verify that everything is still in order. Every block is timestamped, so it is almost impossible to tamper with the data. When a new block is created, it is sent to all the nodes of the chain. Every node verifies that this block has not been tampered with and creates a consensus.

There is no central authority in a blockchain, so it is a decentralized structure. There are two types of blockchain. One is public, e.g., Bitcoin and Ethereum, and the other one is private, which are made specially for some organizations. The self-executable scripts are called smart contracts. These smart contracts are beneficial in terms of avoiding fraud.

### 3.1. Minting

Before adding a new block into the chain, the blockchain ensures the minting process. In this process, different algorithms are used. Every block is distributed over the chain to verify it. Different miners use different algorithms to verify the block. After verification, that block is added to the chain.

### 3.2. Immutability

Immutability is one of the main features of BCT. It protects the chain from being corrupted. The data of the chain remains permanent and cannot be altered. The hashing and encryption functions of the blockchain are very complex, which makes it almost impossible to change or reverse the data over it. If a hacker wants to change the transaction data, s/he has to change that data on every single node of the network, which is impossible.

### 3.3. Reliability

BCT has eliminated the role of the third party and introduced the concept of smart contracts, which allows users to communicate and make transactions independently. The BCT will ensure integrity on the basis of self-executing smart contracts.

### 3.4. Anonymity

One of the exciting features of a blockchain is its anonymity. Although all nodes have a copy of the whole chain, user data remain anonymous. One needs only to see the transaction history if the user has a specific address.

### 3.5. Decentralization

BCT is based on distributed ledger technology, so it is decentralized. There is no governing authority that controls the blockchain. It gives every user full control over his property. The user does not have to rely on any third party to maintain his or her property.

### 3.6. Transparency

Every node of a blockchain has a copy of the whole chain. This feature makes the blockchain transparent, which runs on smart algorithms. Every transaction is viewable and transparent. A hyperledger fabric is a private blockchain platform that embraces all the features required for safe and secure transaction ledgers. Figure 2 depicts the general architecture of a blockchain. The client can interact with distributed ledger technology using REST API. Certificate authority assigns the digital certificates to nodes and peers, and it interacts with the software development kit (SDK) of certificate authority (CA).The client communicates with the client SDK, which generates a request to create a channel. Every channel consists of the membership service provider (MSP), consensus services, and smart contract services. The network admin can designate different roles to different peers using the MSP. Channels also contain a secure image file repository and a container execution service. Channels manage P2P protocols and the endorsement verification module and initiate a transaction to fabric peers. Fabric peers consist of endorsing peers, non-endorsing peers, and orderer peers. Since mining algorithms are not used in a hyperledger fabric, endorsing peers do not act as miners, and orderer peers therefore deliver the blocks. Endorsing and non-endorsing peers interact using gossip protocols.

## 4. A Blockchain-Based Secure Image Encryption Scheme

Image sensors are of great importance in an IIoT-oriented network computing system. In this system, data are acquired through different devices and in different formats according to the requirement. Data are then stored in an internal memory of a sensor or sometimes offloaded to the cloud. Image processing can be done on clouds in weak real-time conditions [32], and further processing is done. Due to different attacks, industries could face many problems, such as the loss of important data, the loss of control of the device disruption of service providers, criminal activity, and the expenses of recovering lost data. IoT devices exchange a large amount of data for processing and analysis with servers both at the edge and at the cloud. When it comes to devices that collect high amounts of data, such as environmental data, we can collect that data and, after encryption, send it to the processing device. The collected information can be hashed. Hashing is creating a fingerprint. If there is a slight change in the information, the whole fingerprint will be changed. That fingerprint can be stored in the blockchain, and we can then compare it with input fingerprint to see if the data has been tampered with or not. This will protect the data from hackers before reaching the destination. Blockchain is a substantially useful ledger when it comes to ensuring the quality of the data. BCT can create controls for a vast number of decentralized industrial devices. It can enable peer-to-peer communications between globally decentralized devices. It could also provide compliance and governance for all autonomous systems while addressing the security complexities of the new landscape that is emerging.

### 4.1. Assumptions

In the IIoT images, sensors play a vital role in making many decisions. Hence, it is assumed that images are sensitive data, and we propose our solution to secure the use of this data. To ensure this, we will encrypt the data and store the hashed value over the tamper-proof ledger. In our discussion, we made assumptions that our output is a decrypted image, which in ideal conditions would be the plain image. We also assume that the size of the original image is m × n, and we can divide the output by 16. Our empirical research illustrates that we can efficiently generate an auxiliary image of 256×256 pixels by considering all the pixels as randomly distributed variables. Since our primary focus is on providing the security system for image encryption, we included the encrypted images and verify their strength using different established tests in the Results section.

### 4.2. Image Encryption

In the image encryption process, an image is converted into a sequence of bytes so that its original shape cannot be accessed. It is a useful technique to protect the contents of digital images [33]. Different cryptographic algorithms perform the encryption process. Ciphered bytes can then be transferred to another system, where it is modified to obtain original values using the decryption process. For both encryption and decryption processes, we use algorithms based on some key.
(1)$$E_{key}\left(X\right)=Y$$
(2)$$D_{key}\left(Y\right)=X$$
Encrypted plain text and its decryption are explained in Equations (Equation 1) and (Equation 2), where $E_{key}\left(X\right)$ shows the encryption function performed on image *X*, and its resulted image is image *Y*. We perform the $D_{key}\left(Y\right)$ function on the encrypted image to retrieve the original image X.

### 4.3. Proposed Encryption Algorithm

In industry, we can easily deploy a permissioned ledger or a private blockchain. A hyperledger fabric is a platform based on a blockchain and fulfills all the requirements for the IIoT environment. It is an open-source platform under the umbrella of Linux foundations [34]. The hyperledger fabric is used in many fields, including medicine [35], supply chains [36,37], logistics [38], and large-scale IoT data [39]. The blockchain helps the IIoT in many problems, including security, trust, and decentralization [40]. A blockchain makes it easy to connect to many devices. Devices could be end nodes or supercomputers that process data. Nodes transfer the cryptographically secure data to other peers so that no hacker can do a man-in-the-middle attack or penetrate it. We can send data to a central location or the decentralized devices connecting with other devices.

The hash in the blockchain is the unique fingerprint of each datum in the chain. A hash that represents every transaction inside a block is known as a Merkel root. Pairs of the hash are repeatedly hashed within a block, over and over, to obtain a single hash value. Figure 3 summarizes the core components of the blockchain system. Any server computer or user can be a node. A node can initiate a transaction, which is the smallest building block of the whole chain. Those transactions combine to make one block, but this block is added to the chain after verification from miners or endorser nodes. There are different sets of rules set by the system administrator to carry out blockchain operations.

In the IIoT system, we propose a private blockchain system. In this system network, the admin can configure the endorser and non-endorser peers. Since the sensor nodes do not have many power resources to run the mining algorithm, some nodes act as validator nodes. These nodes reach a consensus to add a new block to the chain. The responsibility of a node in a network is to sustain the replica of a blockchain. A node or peer is also responsible for processing the transaction. In an IIoT network, nodes are battery- or electricity-powered devices who perform communication and data collection. Figure 4 represents the Whitebox view of a blockchain node in the IIoT network. The node in the blockchain incorporates different blocks, state databases, policies, and a smart contract. The state of the ledger at given factors and times is represented and stored in the state database. The sample Whitebox illustration of a node indicates the relation between the state database and the blocks.

A Merkel root can be used to trace the original block and reconstruct the entire set of transactions. A nonce is an arbitrary number that can be only used once. While creating a hash for a block, the system requests a particular hash value that starts with a certain amount of zeros. These extra constraints make the mixture more difficult to find. To find that value, we combined all the data with a nonce, and we tried to create hash values. The computer estimates the nonce value over and over again until it finds a hash that meets the constraints.

In the permissioned blockchain, some nodes have been declared as endorser nodes by the system administrator. Those peers can verify the transaction. Figure 5 shows the flow of image encryption in the IIoT. It shows how image sensors will transmit data to the blockchain and, after endorsement by specific nodes, how it will become part of the private blockchain. It starts with the image sensors, which could be any device whose job is to capture images. The images will be transferred to some of the nodes connected with the blockchain. Every node has its own private key issued by the certificate authority. The images will be encrypted using the proposed algorithm and then sent to the endorser nodes. After validation by those nodes, this will become part of the chain. The end-user can access the required images by using the hashed transaction ID of a specific image. Table 2 provides the notations used in the simulation code.

Figure 6 explains the transaction sequence that occurs when a new transaction is initiated over the blockchain network. The process of the transaction starts with submitting the proposal from the client. It also initializes the communication over the client SDK. The transaction process uses two different types of peers. One is an endorser peer, and the other is a committer peer. Endorser peers are responsible for simulating and signing the transaction proposals. They can also grant or deny approval. Committer peers can validate transaction results before writing a block of transactions to the distributed ledger.

The certificate authority (CA) assigns the credentials to the clients. These credentials are required by the client application to obtain permission to submit a new transaction. To initiate a new transaction, client applications send transaction proposals to peers to read or update the ledger. The endorser peer receives the simulated results as a particular read and write dataset (RW-set). The response from endorser peers not only includes a RW-set but also a response value. Read data capture the latest current state, and write data hold data that will be written to the world state while executing the transaction. After receiving the response and signing the transaction, the client broadcasts it to the orderer. The orderer orders the transaction into a block and passes the blocks to all committer peers. They will read it and then validate the endorsement policy. Once it is verified, they will perform the write operation. In the end, the committer peer generates an asynchronous notification about the status of the transaction. The orderer peer notifies the other nodes of whether the transaction is successful or not. Once a block records the transaction, it cannot be tampered with [41].

Figure 7 shows a simplified transaction structure. Transactions in every block consists of a transaction header, a signature, a proposal, a response, and a list of endorsements. The header consists of essential metadata about the transaction that might contain the chain code and its version. A signature is used to check that the transaction details have not tampered with. The proposal within the transaction is the request by the client to read or update the ledger. The response is the answer of the endorser peer, which includes a response value and an RW-set. The client application creates cryptographic data that require the application’s private key to generate.

### 4.4. Smart Contract

An electronic copy of the predefined set of rules between two parties in the form of executable code is called a smart contract. Before recording a transaction, the application invokes the smart contract. Figure 8 illustrates the system diagram of the smart contract and its interaction with peer nodes. Input parameters supplied by an application to the smart contract are encoded in the proposal. The input parameters determine the new world state. The response contains all the values of the world state as a RW-set. This set is the output of a smart contract, and if the transaction is successfully validated, it will be applied to the ledger to update the world state. The endorsement is a list of signed transaction responses.

Algorithm 1 describes the steps to secure the image in IIoT. The process starts with the initialization of the web service of the blockchain for nodes of the network. There are many image capturing devices, and each device acts as a node. When a node captures the image and sends it to the chain for processing, the proposed algorithm will perform initial checks. It will verify that the current time is less than that of the message distribution phase and whether the node is registered or not registered. The CA assigns a digital identity to every node of the network. If the node has a cryptographically validated digital certificate, mapped by the CA, then it can participate in the system as shown in Figure 6. If the requested transaction is already processed, then it is ignored. After initial checks, it will start the encryption process for the image. This image is processed, and a hashed transaction ID will be allocated for each image, which is the key of the entire scheme. We use Algorithm 2 for the encryption of the image. This uses an image as message and transforms it into messagedigest. The first step is prepossessing in which a padded message with the same size as the image is created. The image is then parsed into blocks of 512 bits. This gives an output of a 256 bit or 32 byte string of ASCII characters. The hashed key of the blockchain is so secure that it is almost impossible to break. After encrypting the image into a secure string, it will be sent to the chain where all nodes will verify it. After verification, this block will be added to the chain. From this chain, any computing device can obtain this image using the public key of the block.
**Algorithm 1** The blockchain-based image encryption process.**Require:** BlockchainWebService**Ensure:** genesisblock **while** T has not expired **do**  **if** node $N_{i}$ is authenticated == true **then**   **if** request Ri matched == true **then**    **if**
Ri is identified as processed request == false **then**     process for the response to $C_{i}$     Hash(image)     Update chain    **else**     Response to $N_{i}$ that the Ri is not valid    **end if**   **else**    Deny the Request   **end if**   Validate and Add block into chain  **end if** **end while**
**Algorithm 2** Image encryption.**Require:** genesisblock(Gb)**Ensure:** Image(P) Get m x n from P initialize y=uint8(zeros(mxn)) initialize K = 1 sh=rand(1,512×512) [t,Ind]=sort(sh); **while** i ≤ m **do**  **while** i ≤ n **do**   temp=x(i,j:j+31);   y(i,j:j+31)=(Gb(k,:)⊕p(i,j:j+31));   Gb(k+1,:)=uint8(sha256hasher.ComputeHash(y(i,j:j+31)));   increment k by 1  **end while** **end while**

The computing node could be a server, an edge, or the cloud, as shown in Figure 5. To use the images, there are two types of keys. One is a private key obtained by a digital sign 256 bit algorithm: the elliptic curve digital signature algorithm (ECDSA). Another one is a public key, which can be used to share the image with the cloud or other nodes [42]. This key can be obtained by applying the RACE Integrity Primitives Evaluation Message Digest (RIPEMD) algorithm on SHA 256. It will give an output of 160 bits. To shorten this public key, an algorithm called Base58check is used, which gives an output of 58 alphanumeric characters. This durable, hashed encryption of images ensures security and privacy. Since this algorithm proposes the use of two different keys, this process is very unique. The encryption key is for the public, and the decryption key is secret [43].

## 5. Experimental Results and Discussion

By using the simulation of the proposed scheme, we obtain fine results. For experimental purposes, we used MATLAB 2015a on Microsoft Windows 10. The RAM for our system is 8 GB, and our computer is equipped with a Core i7 processor.

We selected four test images ($P_{o}$), shown in Figure 9. These histogram plots ($H_{o}$) in Figure 10 show the variation in the color intensity before any experiment is done on original images. The x-axis of the histogram shows the intensity value. We used grayscale images, so they range between 0 and 255. The y-axis represents the number of pixels containing those intensity values. After applying the blockchain-based encryption scheme on our test images, we obtained encrypted images ($P_{e}$) shown in Figure 11. Figure 12 shows the respective histogram plots ($H_{e}$) of encrypted images.

These plots and images show that the information on these images is completely hidden and unreadable. Now our images are safe and secure. We can offload those images to the cloud for further processing without the fear of any misuse of these images.

We evaluate the security of the proposed scheme using information entropy analysis, the correlation coefficient, the unified average change intensity (UACI), the number of pixels change rate (NPCR), histogram analysis, and noise attack.

### 5.1. Key Security Analysis

The security of a cryptosystem should rely on the secrecy of the key. To analyze the security of encryption algorithms, many systems are in use. Chen’s system [44] is one of them, which is sensitive to the parameters of the system and their initial values. By applying this rule, it will be apparent that the decrypted image will be different from the input image. If the initial values have a slight difference, then it will not allow the image to reshape into its original form. We have done some experiments and simulations to test key sensitivity. The first image of Figure 13a shows the original image, which was used as input. It is a picture of a cameraman, and the second image, Figure 13b, is an encrypted image of this input image. In addition, the last image, Figure 13c, shows the decrypted image obtained using the wrong key. From the pictures, it is clear that the decrypted image is entirely different from the original image, and it is not easy to distinguish any information from the original image. Based on the above argument, our algorithm is sensitive to the secret keys, which demonstrates that it can resist exhaustive attack.

### 5.2. NPCR and UACI Test

A good algorithm should be robust against all different kinds of attacks. In this section, we describe the results of varying performance analyses for the proposed algorithm. We applied the NPCR test and the UACI test. The results of the security analysis show that our proposed algorithm is feasible and effective.
(3)$$L(i,j)=\left\{\begin{array}{cc} 0 & I_{1}(i,j)=I_{2}(i,j)\\ 1 & I_{1}(i,j)\neq I_{2}(i,j) \end{array}\right.$$
(4)$$NCPR=\frac{\sum_{i=1}^{p}\sum_{j=1}^{q}c\left(i,j\right)}{p\times q}\times 100\%$$
(5)$$NCPR=\frac{\sum_{i=1}^{p}\sum_{j=1}^{q}I_{1}\left(i,j\right)-I_{2}\left(i,j\right)}{p\times q\times 255}\times 100\%$$

We refer to the encrypted image as “image1 $\left(I_{1}\right)$”, and the encrypted image obtained after changing the first-pixel gray value from $I_{1}$ is called “image2 $\left(I_{2}\right)$”. We use the NPCR and UACI as criteria to check the resistance performance differential against attack. Thus, we used Equations (Equation 3)–(Equation 5) to calculate two security analysis tests; one is NPCR, and the other is UACI, between $I_{1}$ and $I_{2}$ where p and q are the height and width of the image, respectively, and $I_{1}\left(i,j\right)$ and $I_{2}\left(i,j\right)$ denote the pixel values for “image1” and “image2” at specific point (i,j).

We obtained the following results for our image: NPCR = 99.6023% and UACI = 33.4187%. From the results of our simulations, it is clear that our proposed algorithm has a high capacity for resisting differential attack. Table 3 shows a comparison of our proposed algorithms and other existing algorithms. We used these theoretical values to analyze our NPCR and UACI results. A previous study established a mathematical model for ideally encrypted images and for deriving the expectations and variances in the NPCR and UACI tests [45]. Table 3 shows the test results for these tests on our sample image. Our encrypted image agreed with all of the theoretical values. We compared the results obtained using the proposed algorithm.

### 5.3. Correlation Coefficient Analysis

One of the most established analyses of encryption is done by correlation coefficient analysis. For testing purposes, we continued using the cameraman’s image.

First, we took the correlation coefficient of the horizontal direction of the adjacent pixels of the original image. We then compared it with the correlation coefficient of the different direction of the adjacent pixels of the encrypted image. Figure 14, Figure 15 and Figure 16 depict the correlation coefficient of the original input image of the cameraman and the blockchain-based encrypted image. These two show that the correlation coefficient is very high in the input image, whereas the zero correlation coefficient is satisfied in the hashed image. This demonstrates a tight correlation among the pixels in the diagonal, horizontal, and vertical direction of the original image; however, in the encrypted images, this correlation becomes very weak. The values for the original and encrypted images, along with the diagonal, horizontal, and vertical direction of the cameraman image, are given in Table 4.

### 5.4. Entropy Analysis

Entropy analysis is a significant test for analyzing the randomness of information. Entropy is the degree of uncertainty in communication systems, which can be calculated using Equation (Equation 6). *I* denotes the image, and $p\left(I_{n}\right)$ denotes the probability of $\left(I_{n}\right)$
(6)$$E\left(I\right)=\sum_{n=1}^{256}p\left(I_{n}\right)\log_{2}p\left(I_{n}\right)$$

The maximum entropy value for grayscale images is eight, and we achieve entropy values 7.9972 and 7.9978, which are very close to the ideal value. Table 5 presents the comparison of entropies for Lena and Cameraman images with existing techniques.

### 5.5. Noise Attack

To analyze the robustness of the proposed algorithm, we applied a noise attack on Figure 9. The corresponding decrypted images of Cameraman Figure 17, Lena Figure 18, Man Figure 19, and Truck Figure 20 show the strength of the proposed scheme against noise attack. We used salt and pepper noise with densities ranging from 0.01 to 0.10. Table 6 shows the respective mean-square-errors (MSE) for all images.

## 6. Relevant Issues

In this section, we analyze different issues, which can be faced during the implementation of our proposed system.

### 6.1. Forks and Versioning

A divergence in the blockchain is called a fork. However, in a private blockchain, forks do not occur. A fork happens when the exact block is different at different nodes. In a permission ledger, when a node endorser sees a block at a specific height, it is guaranteed to be a correct block at that height.

### 6.2. Speed

Due to the size of the image, transaction size will be more than simple, and it could affect the transaction per second (TPS) text; in comparison to public blockchains, permissioned or private blockchains are faster [53]. They have a higher TPS. Here, only a few nodes have permission to authorize, unlike public blockchains, where achieving consensus takes time.

### 6.3. Mining

In a private blockchain network for high-cost industry protection, proof of work or proof of cost mining is not required. In an industrial environment, the identity of network participants is known, so the main emphasis is on the overall efficiency of blockchain transactions. Sensors and nodes in IIoT do not have enough power to run energy-hungry mining algorithms. To overcome the need for mining a hyperledger fabric has introduced the concept of validating peers. The network administrator selects these endorser nodes. All validating peers reach a consensus to execute the transaction.

### 6.4. Access Control

A permissioned blockchain is useful in organizations where information must be stored with the permission of certain members of that organization. The network administrator manages the level of access control. In such a case, admin can also grant access control to read and write. A single organization or a group of organizations manages private blockchains. The network administrator in such an environment plays a vital role in assigning new certificates to peers. There must be some check and balance system by the organization on the admin so that, if he is compromised or attacked, the system must remain working. A hyperledger fabric most commonly employs a permissioned blockchain [34]. Monax [54] and multichain [55] are other examples of permission blockchain.

## 7. Conclusions

Hack-proof cryptography of a blockchain eliminates security risks for IIoT. This technology can also track every connected sensor or node. A secure image encryption scheme for an IIoT-oriented network computing system based on a blockchain will prove helpful in safely offloading data from devices. We carried out several tests to verify that our proposed algorithm is secure. There are still, however, some limitations in the use of this technology, including limited computing resources and the speed of transactions. Many IIoT devices such as connecting sensors are deficient in memory and processing in terms of resources, which prevents them from acting as nodes in a blockchain. Web services can resolve this problem, but this issue still needs to be addressed. In the future, security for images on the cloud after offloading can also be considered. More work will be done on this technology to make it more efficient and adaptable. 

## Figures and Tables

**Figure 1 entropy-22-00175-f001:**
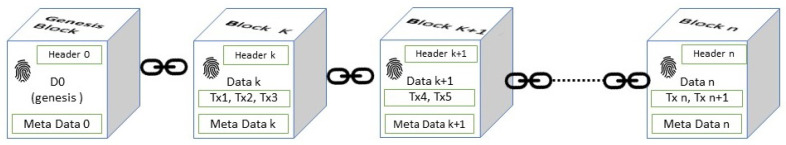
A simplified structure of a blockchain.

**Figure 2 entropy-22-00175-f002:**
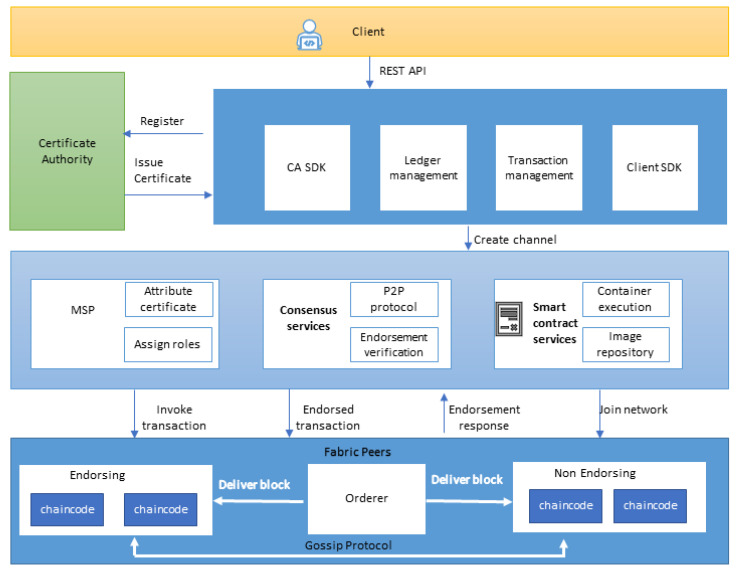
Architecture diagram of a blockchain.

**Figure 3 entropy-22-00175-f003:**
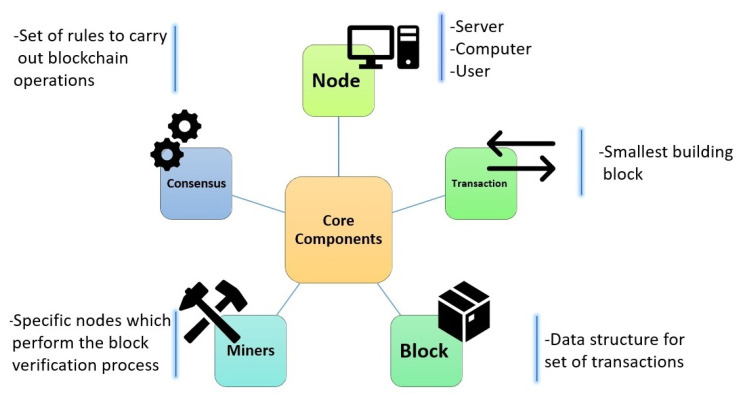
Core components of the blockchain system.

**Figure 4 entropy-22-00175-f004:**
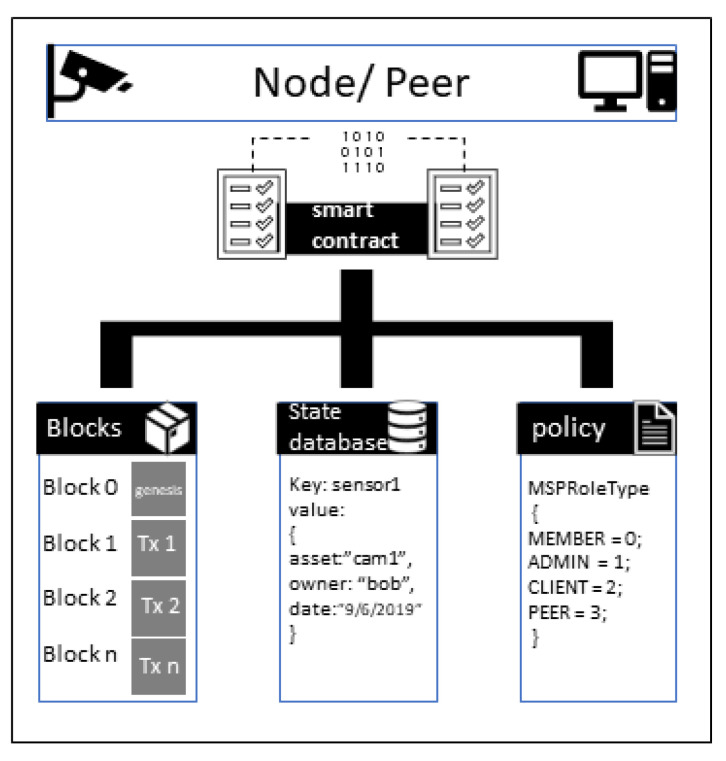
Whitebox view of a blockchain node.

**Figure 5 entropy-22-00175-f005:**
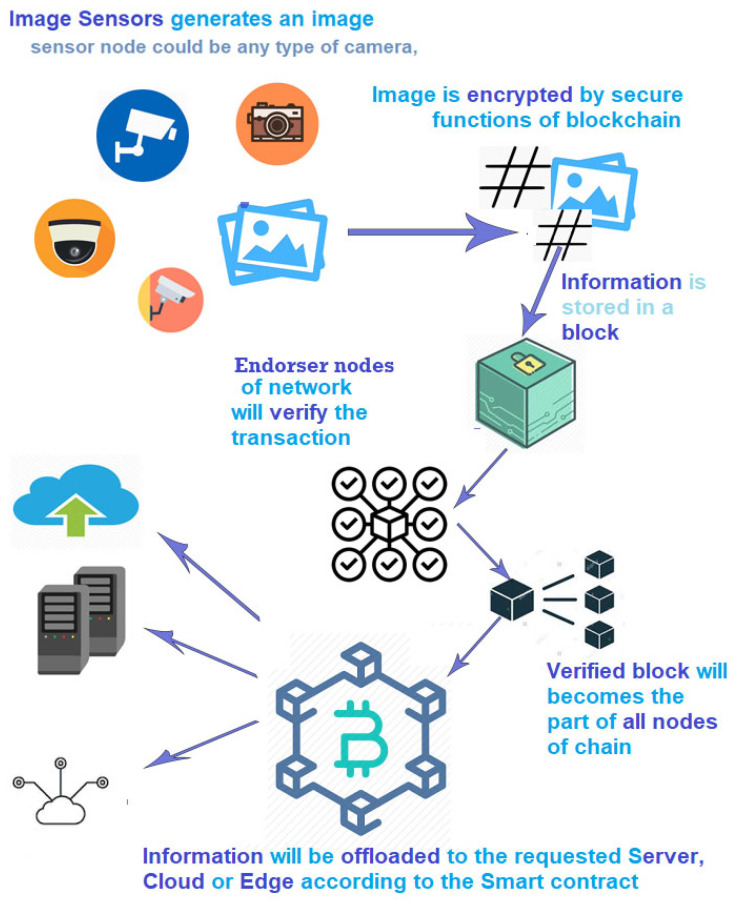
Blockchain-based image encryption system.

**Figure 6 entropy-22-00175-f006:**
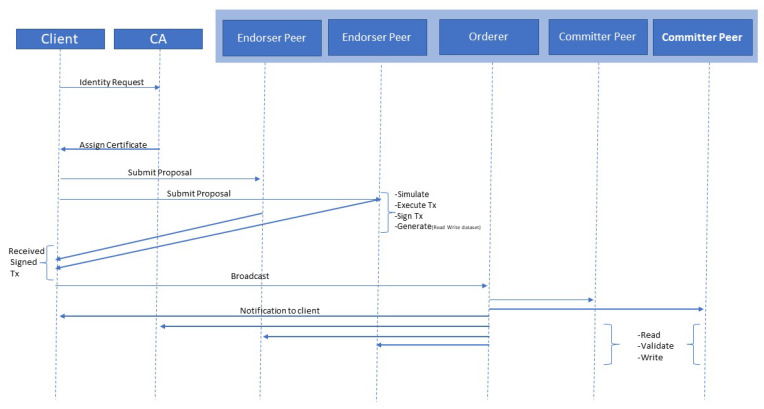
Sequence diagram of transaction operational processes.

**Figure 7 entropy-22-00175-f007:**
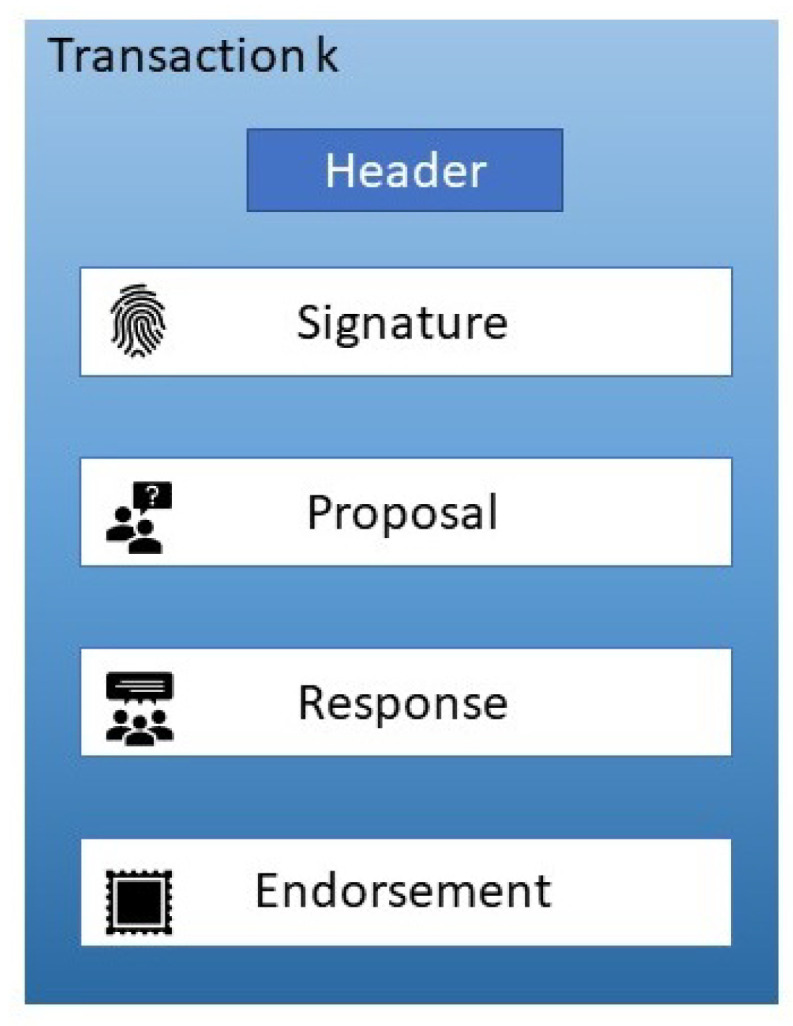
Transaction details.

**Figure 8 entropy-22-00175-f008:**
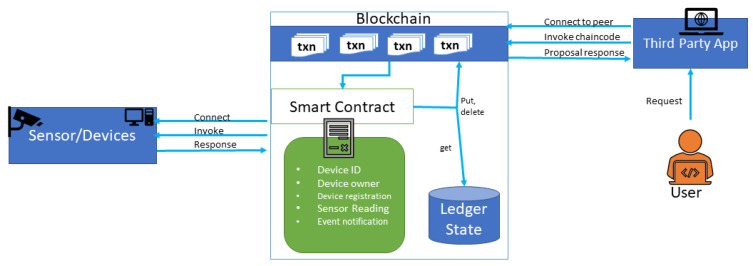
Smart contract in the industrial Internet of Things (IIoT) environment.

**Figure 9 entropy-22-00175-f009:**
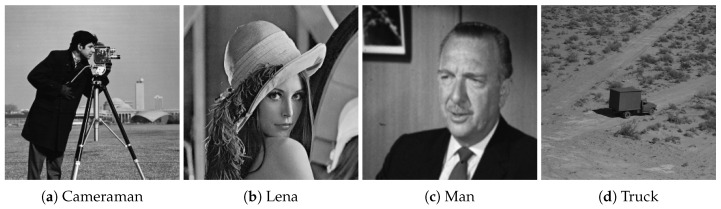
Original images.

**Figure 10 entropy-22-00175-f010:**
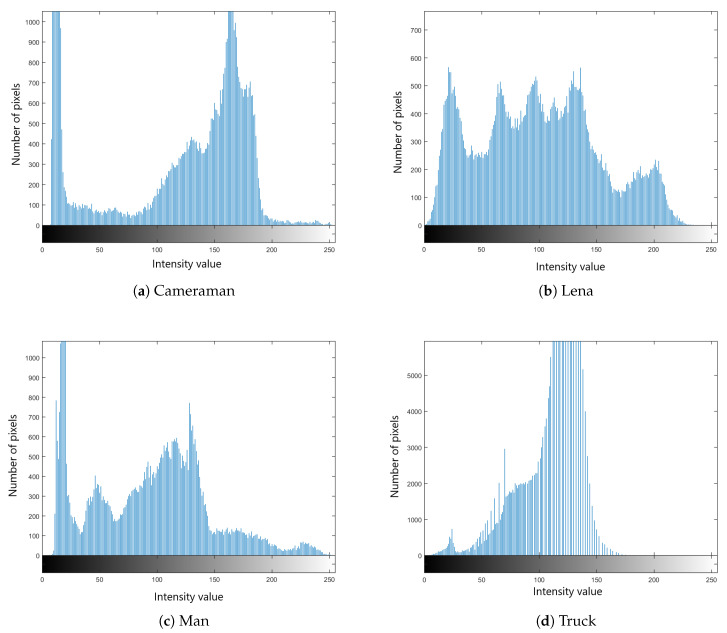
Histograms of original images.

**Figure 11 entropy-22-00175-f011:**

Encrypted images.

**Figure 12 entropy-22-00175-f012:**
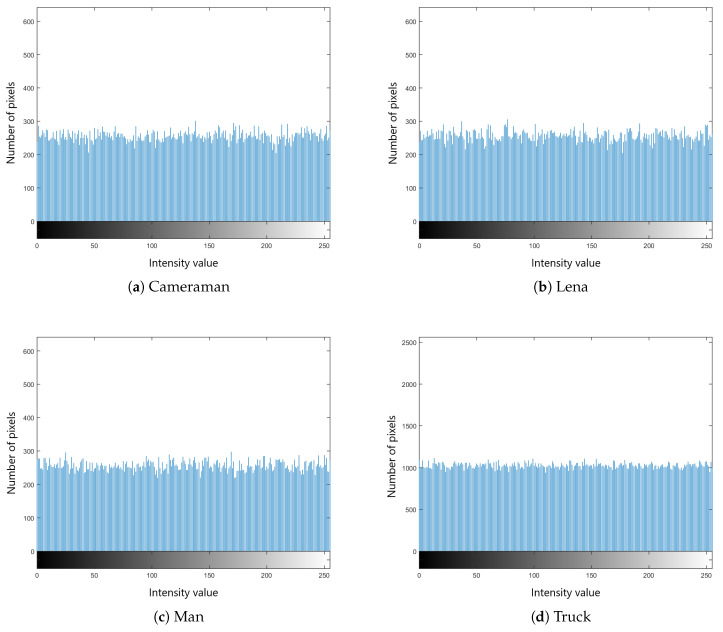
Histograms of encrypted images.

**Figure 13 entropy-22-00175-f013:**
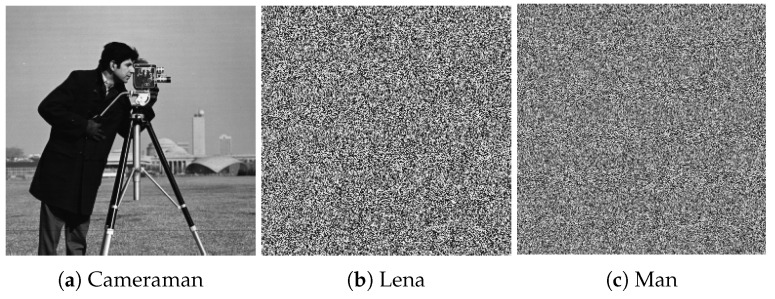
Key security analysis.

**Figure 14 entropy-22-00175-f014:**
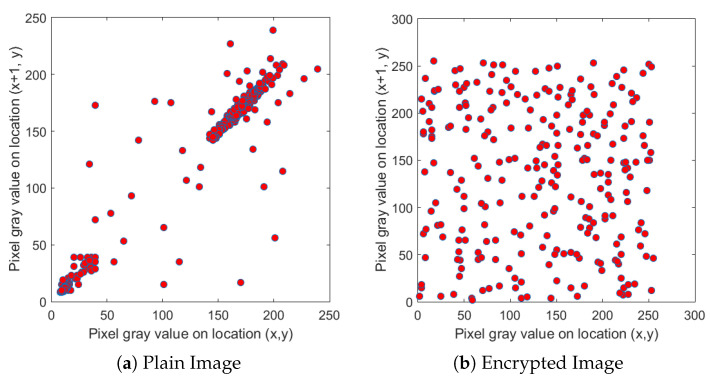
Relationship between neighboring pixels in horizontal directions.

**Figure 15 entropy-22-00175-f015:**
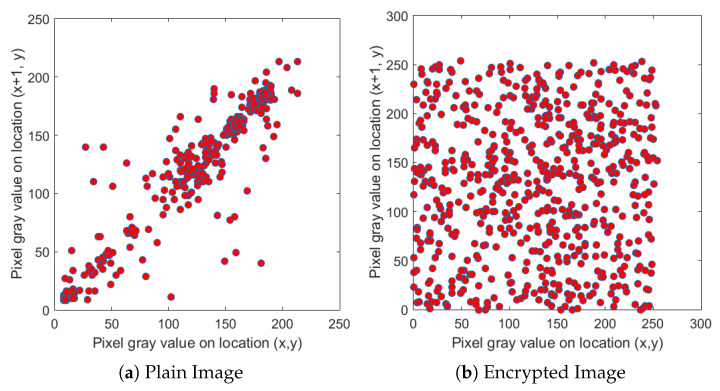
Relationship between neighboring pixels in vertical directions.

**Figure 16 entropy-22-00175-f016:**
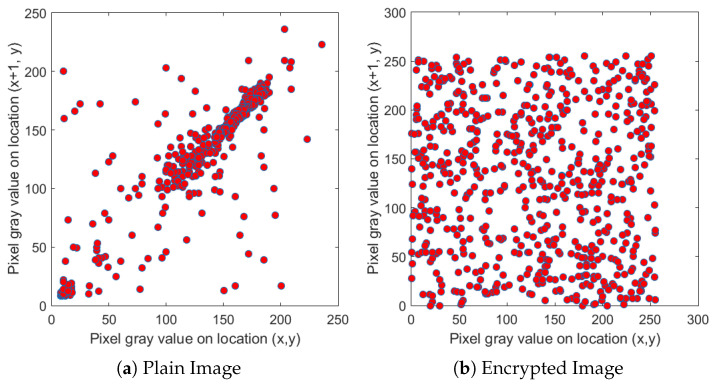
Relationship between neighboring pixels in diagonal directions.

**Figure 17 entropy-22-00175-f017:**
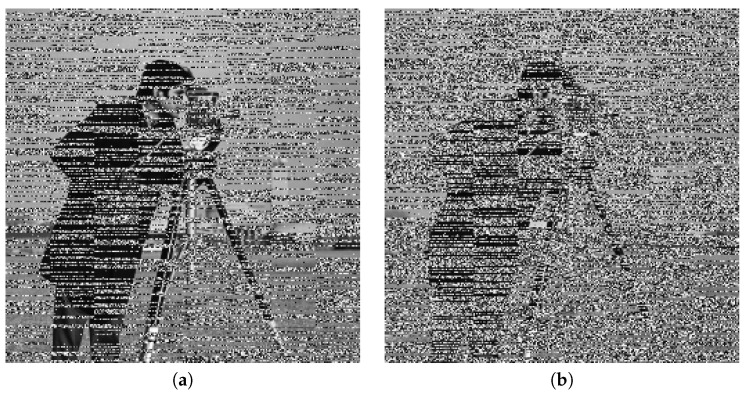
Results of noise attack on the Cameraman image; (**a**) density = 0.02 (**b**) density = 0.04.

**Figure 18 entropy-22-00175-f018:**
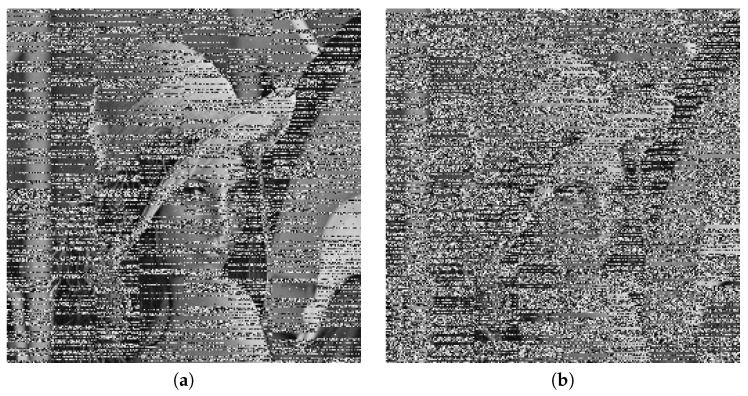
Results of noise attack on the Lena image; (**a**) density = 0.02 (**b**) density = 0.04.

**Figure 19 entropy-22-00175-f019:**
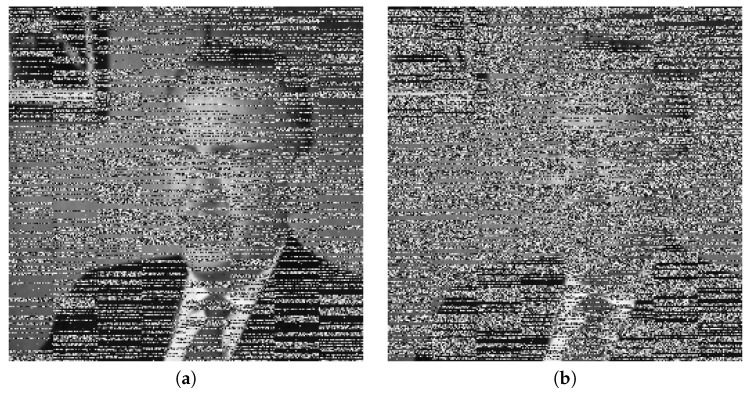
Results of noise attack on the Man image; (**a**) density = 0.02 (**b**) density = 0.04.

**Figure 20 entropy-22-00175-f020:**
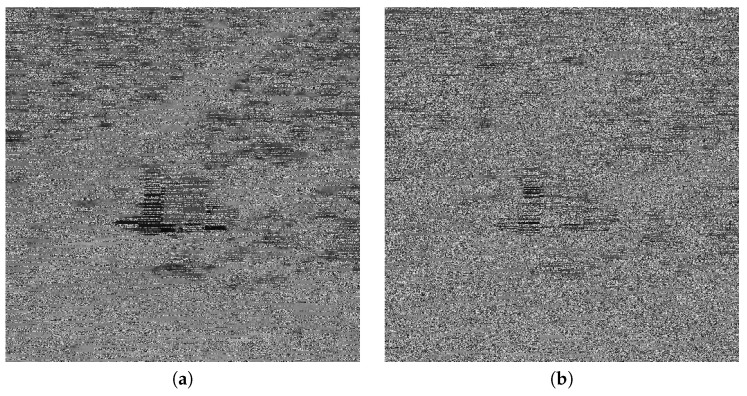
Results of noise attack on the Truck image; (**a**) density = 0.02 (**b**) density = 0.04.

**Table 1 entropy-22-00175-t001:** Comparison of existing encryption solutions.

Sr #	Year	Correlation Coefficient	Encryption Quality	Key Sensitivity	Technique	Decentralized
1 [22]	2017	Presented	Normal	No	2D Cat Map and Shadow method	No
2 [23]	2017	Presented	Normal	No	ACM with Henon and Logistic Map.	No
3 [24]	2017	Not Presented	Normal	No	Scalable Hierarchical model	Yes
4 [25]	2018	Presented	Normal	Yes	Keystream generation strategy	No
5 [26]	2018	Not Presented	Good	Yes	Logistic sine coupling	No
6 [27]	2019	Presented	Normal	Yes	Three dimensional chaotic system	No
7 [28]	2018	Presented	Good	Yes	Linear coupling logistics map	No
8 [29]	2019	Presented	Normal	No	Coupled logistic bernoulli map	No
9 [30]	2019	Presented	Normal	Yes	Orbit Perturbation	No
10 [proposed]	-	Presented	Good	Yes	Blockchain	Yes

**Table 2 entropy-22-00175-t002:** Notations used in experiments.

*Sr No.*	Notation	Description
1	$T_{x}$	Transaction
2	$G_{b}$	Genesis Block
3	$P_{o}$	Original Picture
4	$P_{e}$	Encrypted Picture
5	$sha_{256}$	Hash Algorithm
6	$H_{o}$	Histogram of Original Picture
7	$H_{e}$	Histogram of Encrypted Picture

**Table 3 entropy-22-00175-t003:** Deferential attack comparison.

	Our	Zhou et al. [46]	Belazi et al. [47]	Muhammad et al. [48]
NPCR	99.6023	99.6023	99.6098	99.61
UACI	33.4187	33.45	33.4384	33.44

**Table 4 entropy-22-00175-t004:** Results of correlation analysis.

Model	Original Image	Ciphered Image
HORIZONTAL	0.944198	−0.042225
VERTICAL	0.961276	0.036725
DIAGONAL	0.899276	−0.058265

**Table 5 entropy-22-00175-t005:** Comparison of entropies for the Lena and Cameraman images.

Model	Lena	Cameraman
[49]	7.5755	7.9971
[50]	7.9965	7.9904
[51]	7.9974	7.9971
[52]	7.9832	7.9834
Proposed	7.9978	7.9972

**Table 6 entropy-22-00175-t006:** MSE between the input and the corresponding decrypted images.

Sr #	Density	MSE			
		Lena	Truck	Man	Cameraman
1	0.01	2.58	1.83	2.81	2.62
2	0.02	4.32	3.14	4.85	4.48
3	0.03	5.73	4.14	6.22	5.95
4	0.04	6.72	4.82	7.37	6.92
5	0.05	7.32	5.34	8.13	7.82
6	0.06	7.90	5.72	8.62	8.17
7	0.07	8.31	6.03	9.09	8.56
8	0.08	8.43	6.15	9.32	8.74
9	0.09	8.62	6.29	9.61	9.01
10	0.10	8.69	6.40	9.78	9.14

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
