# Peer review of "A Blockchain-Based Secure Image Encryption Scheme for the Industrial Internet of Things"

_entropy, 2020, doi:10.3390/e22020175_

Round 1
Reviewer 1 Report
The paper undertook a major revision mainly related to "Literature Discussion", where the contribution of work was highlighted. However, the section related to Results continues without a proper presentation. Results are presented in a very superficial way, without an in-depth comparison with algorithms presented in previous sections. Reviewers described other issues in the first round; however, they were not attacked by the authors, as follows:
1) Again, Figure 1 is placed on page 2, and its citation is on page 5. Put the figure as close as the citation (page 4 or 5).
2) Figure 2 is suitable for a presentation, but for a paper is has an exaggerated size. The information described in Figure 2 is not necessary to the understanding of the paper.
3) The sequence diagram presented in Figure 6 is poorly described in the text.
4) Figures 10, 11, 12, and 13 continue with several flaws. No sub captions are indicating who is a), b), c), or d).
5) There are no axis names in Figures 10 and 12.
Author Response
Dear
REVIEWER 1
We would like to thank you for the thoughtful comments and constructive suggestions, which have been helpful to improve the quality of this manuscript. We hope that this revised manuscript meets your expectations.
Regards
Again, Figure 1 is placed on page 2, and its citation is on page 5. Put the figure as close as the citation (page 4 or 5). We agree with the reviewer’s advice and have therefore moved Figure 1 to page 4, close to its citation (Page: 4, Line 144) Figure 2 is suitable for a presentation, but for a paper is has an exaggerated size. The information described in Figure 2 is not necessary to the understanding of the paper. Thank you for pointing this out. We agree with this comment. Therefore, we have removed the Figure 2 (Figure 2. Key features of Blockchain) The sequence diagram presented in Figure 6 is poorly described in the text. We have added more details about the sequence diagram (Line 266-284) Figures 10, 11, 12, and 13 continue with several flaws. No sub captions are indicating who is a), b), c), or d). We have, accordingly, modified Figures 10,11,12 and 13. We have added their Sub captions. (Page: 14,15,16) There are no axis names in Figures 10 and 12. Thank you for pointing this out. The suggested correction has been made. Axis names in Figures 10 and 12 are added (Page: 14,15)

Reviewer 2 Report
The authors provided answers to the reviewer's comments but open more questions. For example, the use of Hyperledger Fabric is considered but this solution demands from the involved partners to set up the infrastructure (therefore details regarding the nodes should be provided). In literature the most well-known example of using Hyperledger Fabric is IBM's solution for the supply chain but this solution is backed up by a large organisation that could maintain and set up the infrastructure which is not the case for every company.
Author Response
Dear
REVIEWER 2
We would like to thank you for the thoughtful comments and constructive suggestions, which have been helpful to improve the quality of this manuscript. We hope that this revised manuscript meets your expectations.
Regards
The authors provided answers to the reviewer's comments but open more questions. For example, the use of Hyperledger Fabric is considered but this solution demands from the involved partners to set up the infrastructure (therefore details regarding the nodes should be provided). As suggested by the reviewer, the modification has been done in the updated manuscript. We have described the details of node on page (Line 243-249) In literature the most well-known example of using Hyperledger Fabric is IBM's solution for the supply chain but this solution is backed up by a large organisation that could maintain and set up the infrastructure which is not the case for every company. You have raised an important point here. However, Hyperledger Fabric is an open-source platform under the umbrella of Linux foundations. We have included the use cases of Hyperledger fabric along with recently published articles. In the revised manuscript, we modified page 7, where we mentioned the use of Hyperledger Fabric in many fields including medical, drugs supply chain IoT networks and in industrial IoT. (Page: 4, Line 229-231)

Reviewer 3 Report
After careful consideration, I found an effort from authors to satisfy my minor comments with some improvements.The manuscript is fine to me for possible publication.
Author Response
Reviewer 3
After careful consideration, I found an effort from authors to satisfy my minor comments with some improvements. The manuscript is fine to me for possible publication.We would like to thank you for the thoughtful comments and constructive suggestions, which have been helpful to improve the quality of this manuscript. We hope that this revised manuscript meets your expectations.

Round 2
Reviewer 1 Report
The authors revised the paper according to reviewers' comments.
Author Response
The authors revised the paper according to reviewers' comments. We would like to thank you for the thoughtful comments and constructive suggestions, which have been helpful to improve the quality of this manuscript.
Reviewer 2 Report
The authors have responded to the previous comments. I believe that they should include a paragraph emphasizing on the benefits of the Blockchain in the topic.
Author Response
The authors have responded to the previous comments. I believe that they should include a paragraph emphasizing on the benefits of the Blockchain in the topic. As suggested by the reviewer, we have added a new paragraph to emphasize the benefits of blockchain in our topic (Line 136-154)
This manuscript is a resubmission of an earlier submission. The following is a list of the peer review reports and author responses from that submission.
Round 1
Reviewer 1 Report
The authors propose the application of a blockchain approach for security context when considering image encryption in the Industrial Internet of Things. The idea sounds scientific and very trendy; however, some points should be highlighted:
Typos and syntax errors are present overall text. A revision considering language should rbe taken. Even though Section 1 is very well written, the existing gaps regarding the use (in IIoT context) of the blockchain approach in image encryption are missing. What is open in the literature about that? The position of Figure 1 is out of context. Section 2 must undertake a major revision in which the literature review must focus only on the target topic of the paper. Only a little works were cited. The articles cited in Table 1 should be doubled, considering the vast amount of papers in this area. Table 1 must be part of Section 2. It would be enlightening if a comparison with the proposal is present in Table 1. Section 3 is short. A more didact explanation should be introduced about theoretical issues presented in the topic handled by the paper. A subsection "assumptions" should be introduced in Section 4. A lot of information is taken on before section 4.1. Formal presentation (mathematical equations) is missing in Section 4. Details of implementations presented in Figs. 3 and 4 were not described in the paper. This part is mandatory because those figures are the core of the proposal. Thus, without that information is hard and very difficult to reproduce the proposal. A formal presentation of Algorithm 1 is missing. One of the main problems in the text is that it is not very easy to relate results (Section 5) with the proposal (Section 4). The fonts of axes present in Figs.8 and 10 must be increased. It is not possible to read them. Fig. 9 is out of the context. It is mandatory to relate the results with models presented in the previous section to clarify the contributions of the paper. The results presented in Tables 3 and 5 are almost the same as those found in the literature.
Reviewer 2 Report
The authors propose the use of a Blockchain network as a secure image encryption scheme for IIoT. Unfortunately, the part that is focused on Blockchain is not at all covered in depth. The authors write that they will use a private Blockchain but they do not mention which platform they have in mind. At the same time, they are not providing details for the consensus algorithm that would be used (will it be Proof or Work (PoW) or not?) and, therefore, their statement that forks cannot happen in private Blockchain cannot be proved. On top of this, details about the Blockchain network are not provided. For example, which devices will be the nodes? Will the IoT devices have to be miners and, then, possibly run the energy hungry PoW algorithm?
In summary, the Blockchain part seems to be lacking much details in order to be considered as a viable solution to the problem while it is your main argument for the efficiency of your algorithm.
Reviewer 3 Report
The authors are suggested to revised the manuscript based on the following comments.
Smart contracts is on of the key contribution of your work. Then, operations of smart contracts for image encryption and data interactions must be elaborated. Mining process for blockchain is completely ignored. Please elaborate it in the blockchain process. Algorithm 1: Please elaborate how the Node N(i) can be authenticated? Security for data on cloud is not considered. You know, offloading data and storing it on cloud may not ensure security. Please perform a formal analysis on that. You stated “The level of access control is managed by the network administrator”. What problem happens when the administrator is compromised or attacked? Solutions for this issue should be clarified.Reviewer 4 Report
Overall, the manuscript is interesting and attractive. The capability of the encryption cryptosystem based on the blockchain has been demonstrated in the paper by using NPCR, UACI and entropy analysis. Here are some comments:
1: In sectoin 4.2, the authors should Clarify the encryption algorithm. Since I did not find the specific algorithm for encryption process.
2: In section 5, apart from the existing analysis, I recomend the authors consider some potential attack experiments. Like occluded attack, noise attack...
3: Some latest articles you might read and cited:
Huang L, Cai S, Xiong X, et al. On symmetric color image encryption system with permutation-diffusion simultaneous operation[J]. Optics and Lasers in Engineering, 2019, 115: 7-20. Chen H, Liu Z, Zhu L, et al. Asymmetric color cryptosystem using chaotic Ushiki map and equal modulus decomposition in fractional Fourier transform domains[J]. Optics and Lasers in Engineering, 2019, 112: 7-15. He X, Tao H, Liu C, et al. Single-shot color image encryption based on mixed state diffractive imaging[J]. Optics and Lasers in Engineering, 2018, 107: 112-118. Chen H, Du X, Liu Z. Optical hyperspectral data encryption in spectrum domain by using 3D Arnold and gyrator transforms[J]. Spectroscopy Letters, 2016, 49(2): 103-107. Zhu K, Lin Z, Ding Y. A New RSA Image Encryption Algorithm Based on Singular Value Decomposition[J]. International Journal of Pattern Recognition and Artificial Intelligence, 2019, 33(01): 1954002.